

# Evaluating an Earth system model from a water user perspective

Mari R. Tye[1,2], Ming Ge[1], Jadwiga H. Richter[1], Ethan D. Gutmann[1], Allyson Rugg[1], Cindy L. Bruyère[1], Sue Ellen Haupt[1], Flavio Lehner[3,1,4], Rachel McCrary[1], Andrew J. Newman[1], Andy Wood[1,5]

1 National Center for Atmospheric Research, Boulder, CO
2 Whiting School of Civil Engineering, Johns Hopkins, Baltimore, MD, USA
3 Department of Earth and Atmospheric Sciences, Cornell University, Ithaca, NY, USA
4 Polar Bears International, Bozeman, MT, USA
5 Department of Civil and Environmental Engineering, Colorado School of Mines, Golden, CO, USA

*Correspondence to*: Mari R. Tye (maritye@ucar.edu)

**Abstract**

The large spatial scale of global Earth system models (ESM) is often cited as an obstacle to using the output by water resource managers in localized decisions. Recent advances in computing have improved the fidelity of hydrological responses in ESMs through increased connectivity between model components. However, the models are seldom evaluated for their ability to reproduce metrics that are important for practitioners, or present the results in a manner that resonates with the users. We draw on the combined experience of the author team and stakeholder workshop participants to identify salient water resource metrics and evaluate whether they are credibly reproduced over the conterminous U.S. by the Community Earth System Model v2 Large Ensemble (CESM2). We find that while the exact values may not match observations, aspects such as interannual variability can be reproduced by CESM2 for the mean wet day precipitation and length of dry spells. CESM2 also captures the proportion of annual total precipitation that derives from the heaviest rain days in watersheds that are not snow-dominated. Aggregating the 7-day mean daily runoff to the watershed scale also shows rain-dominated regions capture the timing and interannual variability in annual maximum and minimum flows. We conclude there is potential for far greater use of large ensemble ESMs, such as CESM2, in long-range water resource decisions to supplement high resolution regional projections.

## 1        Introduction

Water availability and water quality for human consumption, ecosystems, and agriculture are fundamental requirements, making pertinent assessments of future change crucial for adaptation planning (IPCC, 2022). Climate related changes in the hydrologic cycle will affect substantial portions of the world population, most directly through changes in water availability at or near the surface (Mankin et al., 2020; Sedláček and Knutti, 2014). The information required by water resource managers for decision making is not readily available in a relevant format, or at sufficient spatial or temporal resolutions from global Earth system models (ESM; e.g., Ekström et al., 2018). We explore how the Community Earth System Model (CESM) represents the climatology




of water availability, focussing on metrics that are familiar to decision makers in planning investment-scale decisions.

The inability of ESMs to explicitly resolve sub-grid scale (~100 km) processes is often cited as the limitation preventing direct model use in decision making. Literature from large organizations making infrastructure

decisions (e.g., Brekke, 2011; Brekke et al., 2009; Reclamation, 2016, 2014) emphasize downscaling climate model data closer to the scale of the watersheds they manage. These additional modeling steps add complexity and may increase statistical errors (Clark et al., 2015; Ekström et al., 2018). Extracting useful and robust information directly from ESMs would reduce such errors if metrics most important to decision makers, such as the timing of peak flow, were known to be robustly represented.


There are many comprehensive examples of metrics used to evaluate climate and hydrological models (e.g., Ekström et al., 2018; Mizukami et al., 2019; Wagener et al., 2022), and communicate the impacts of climate change (e.g., Reed et al., 2022), or to identify decision-relevant metrics (e.g., Bremer et al., 2020; Mach et al., 2020; Underwood et al., 2018; Vano et al., 2014). However, very few have examined whether user defined metrics

can be reliably reproduced by ESMs (Mankin et al., 2020), and if further model development and scale reduction is warranted instead of improved communication (Pacchetti et al., 2021). Better communication may also reduce the temptation of some users to calculate "standard hydroclimate metrics" that are not supported by the climate model data (Ekström et al., 2018).

In contrast, climate model output can be rejected unnecessarily when simulated annual minima from freely running simulations do not "match" the sequence of observed low flows (Ekström et al., 2018; Moise et al., 2015). Similarly, the benefits of a range of projected outcomes from different climate models are not widely appreciated beyond the climate model community (Tebaldi and Knutti, 2007). Large ensembles from a single climate model initialized with a range of atmospheric and ocean conditions, such as the CESM2 Large Ensemble (LENS2;

Rodgers et al., 2021), help to bound the uncertainty that derives from a naturally chaotic system. Averaged over the full ensemble, they give a better estimate of the model's response to internal and external forcing (Deser et al., 2012) and enable assessments of the rarity of projected extremes. The additional analysis to identify structural (i.e. model formulation) and internal variability within regional climate models means that there are fewer large ensembles at a high resolution (Deser et al., 2020).


Since different decision makers have different priorities and time-scales of interest, Shepherd et al. (2018) recommended the development of climate storylines to communicate with those using climate data to make decisions. Informed by prior surveys of water managers (e.g., Brekke, 2011; Brekke et al., 2009; Cantor et al., 2018; Raff et al., 2013; Wood et al., 2021), Fig. 1 aims to map the different types of water decisions (e.g., Raff et

al., 2013 Fig. 3) to the different scales of model resolution (Meehl et al., 2009 Fig. 2). Water managers make daily operational decisions (e.g., to control instantaneous river flow) with the aid of fine-scale weather and flood models (<4 km) that reliably represent convective and local weather scale processes even though their predictability is relatively short lived (Yuan et al., 2019; far left side of Fig. 1). Larger watershed operations (such as reservoir management or groundwater recharge; e.g., Regional Water Authority, 2019) depend on seasonal outlooks



(middle left of Fig. 1). Smaller adaptation and mitigation projects take place at the typical policy or decadal prediction scale (i.e., 4-10 years; middle right of Fig. 1). Finally, major public investments and inter-basin agreements occur at the same time scales as climate projections (30-100 years; far right of Fig. 1) where persistent and relatively predictable synoptic and planetary scale processes are well represented in lower resolution (~100 km) climate models (Phillips et al., 2020). While forecasts (seasonal or decadal) are re-initialized from specific

atmosphere, ocean or land states at regular time intervals, climate projections are run freely from a variety of atmospheric and oceanic conditions that take several decades to converge to a mean climatology. In considering the utility and useability of information directly from ESMs we focus on decisions made over decadal to climate scales at larger spatial scales.

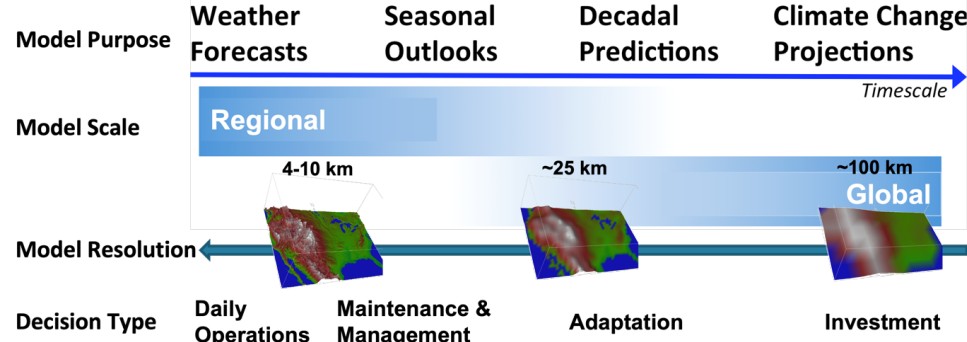


**Figure 1: Mapping the temporal and spatial scales of models to the timeframes for water management decisions.**

Given that ESMs have advanced immeasurably in the recent decade, it is time to re-evaluate whether their direct output can support decision makers. Such an evaluation needs to focus on how well the models can reproduce metrics used by decision makers, and whether the results are credible (Briley et al., 2020; Jagannathan et al.,

2021). Here we evaluate the credibility of one ESM in generating metrics known to be salient for water management decisions; specifically, decisions for water management infrastructure project investments.

The motivation for this paper is to identify:
- a set of water availability metrics that resonates with decision makers and supports their investment-scale

decisions;
- how well CESM2 represents the climatology and recent observed behaviors of those metrics; and
- how such metrics are projected to change.

This paper builds off a decade of collaboration between scientists at the National Center for Atmospheric Research

(NCAR) and US water agencies that lead to a virtual workshop (Tye, 2023), and presents a test case for improved communication with water resources decision makers. The focus is on the Conterminous United States (CONUS) to match the interest of workshop participants.





## 2      Climate Information Needs from Prior Research

Information needs vary greatly, from 5-minute rainfall totals at a point (ASCE, 2006), to basin-wide measures of
annual minimum and maximum total runoff. Water management decision metrics can be grouped into similar
types such as timing, frequency, magnitude, extreme values, variability, and duration of events (Ekström et al.,
2018). While some aspects of timing, magnitude, or variability can be reliably reproduced by ESMs (e.g., Deser
et al., 2020; Tebaldi and Knutti, 2007), others such as short duration extremes are less reliable.

Methods of evaluation and data use also differ. For instance, Clifford et al. (2020) reported that predicting general
changes in the frequency of extreme precipitation events is more useful for future planning than the precise
prediction of mean values evaluated by model developers. Lehner et al. (2019) emphasized that models need to
be evaluated for their ability to reproduce sensitivities (e.g., streamflow changes in response to temperature and
precipitation changes) in addition to mean states. However, metrics that are meaningful for evaluating a model's
capabilities (e.g., the ratio of precipitation to runoff) are less valuable for management decisions (Lehner et al.,
2019; McMillan, 2021; Mizukami et al., 2019). When reporting results, users are more familiar with the 'water
year', rather than the calendar year, to capture the full annual hydrological cycle (Ekström et al., 2018). While the
use of water years is a nuance that does not add substantial value to climate model assessments, communication
with decision makers is improved by presenting data in a familiar format (Briley et al., 2020).


There is a need for information at the local scale that is unlikely to be met directly by raw outputs from the current
generation of ESM. But better communication of the variability in future daily precipitation and associated runoff
can add value to the detailed models by bringing in the added statistical context and perspective of the large
ensembles. Thus, we believe that ESMs can produce useful information about hydro-meteorological extremes
when presented at different spatial or temporal scales, and offer the benefits of large climate model ensembles to
constrain future impact uncertainty.

Appendix A summarizes potential hydrological metrics used in water management decisions (Jagannathan et al.,
2021) or statistical assessments of extremes (Zhang et al., 2011), and model evaluations (Phillips et al., 2020).
Metrics in bold are presented in this paper. We only considered a simplistic measure of meteorological drought
(absence of rain) in the current work, as drought is sensitive to the definition (Bachmair et al., 2016) and local
conditions (Mukherjee et al., 2018), and so not suited to a generalized assessment. Similarly, snow measures are
not included in this assessment. In part due to limited availability of high-quality, long-duration, quality-
controlled, observational data (McCrary et al., 2017); and the biases in snow distribution arising from the
smoothed topography in GCMs (McCrary et al., 2022).

## 3      Data and Methods

### 3.1     Climate Data

CESM2 (Danabasoglu et al., 2020) is a fully coupled global model that simulates the Earth's climate system
through interactive models for atmosphere, ocean, land, sea-ice, river runoff, and land-ice. Variables considered



in this project are taken from the Community Atmosphere Model version 6 (CAM6) and the Community Land Model version 5.0 (CLM5; Lawrence et al., 2019) and are part of the default model outputs. This project uses daily values scaled up to annual (e.g., annual maximum daily precipitation) on a ~1 degree resolution grid. Data were extracted over the CONUS from 10 ensemble members of LENS2 (Rodgers et al., 2021) for model validation in the current era (1981-2010), and a future time period (2041-2070) under the Shared Socioeconomic Pathway

emissions scenario SSP2-4.5 (Riahi et al., 2017). This emissions scenario represents a world where "social, economic, and technological trends do not shift markedly from historical patterns" (O'Neill et al., 2017).

### 3.2    Observations

Gridded daily observations of precipitation at 1/16° horizontal resolution (~6 km) were obtained from the Livneh et al. (2013) dataset covering CONUS and southern Canada for the control period (1981-2010), hereafter referred

to as "Livneh".

Livneh daily temperature maxima and minima, and precipitation were used to force the Variable Infiltration Capacity Model (VIC; Liang et al., 1994) version 4.1.2 to obtain runoff estimates for years 1980-2005 as evaluated in Livneh et al. (2013). Hereafter referred to as "Livneh-VIC".


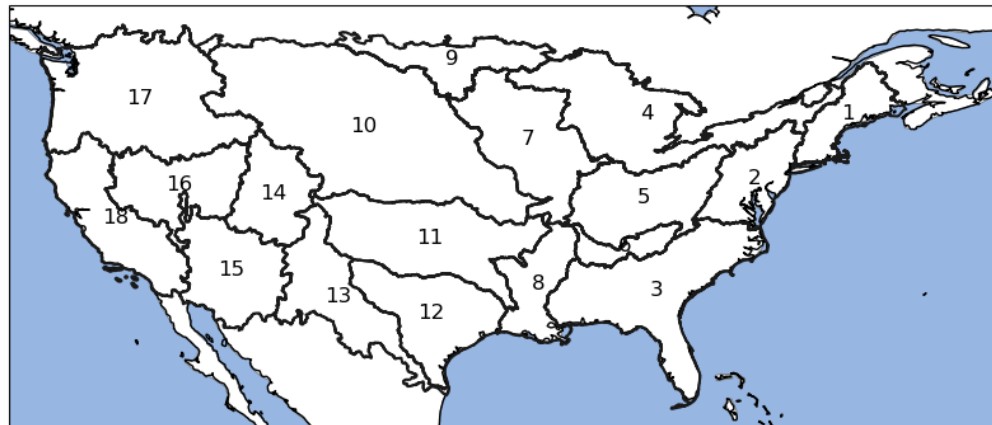

**Figure 2: HUC 2 regions used in data validation and analysis. Regions defined by USGS (2013): Region 01 New England (NE); Region 02 Mid-Atlantic (MA); Region 03 South Atlantic-Gulf (SA); Region 04 Great Lakes (GL); Region 05 Ohio (OH); Region 06 Tennessee (TN); Region 07 Upper Mississippi (UM); Region 08 Lower Mississippi (LM); Region 09**

**Souris-Red-Rainy (RR); Region 10 Missouri (MR); Region 11 Arkansas-White-Red (ARK); Region 12 Texas-Gulf (GUL); Region 13 Rio Grande (RIO); Region 14 Upper Colorado (UC); Region 15 Lower Colorado (LCO); Region 16 Great Basin (GB); Region 17 Pacific Northwest (PN); Region 18 California (CA)**

### 3.3    Methods

All analyses were carried out using the North American water year (1 October to 30 September) to facilitate later communication.



### 3.3.1 Remapping

For ease of comparison, model output were re-gridded using a conservative second-order remapping (Jones, 1999) to place both datasets on the same scale grid and assess anomalies. Data were also calculated as areal averages or totals over the 2-digit Hydrological Unit Code (HUC2) regions (Seaber et al., 1987). HUC2 basins represent 18 watersheds, covering areas ranging from 41,000 mi$^2$ (~105,000 km$^2$; Tennessee) to 520,960mi$^2$ to (1,350,000 km$^2$; Missouri), shown in Fig. 2.

### 3.3.2 Percentile-based thresholds

The threshold for very heavy rain days (Q95) was calculated at each individual grid cell using only days with ≥ 1 mm rain ("wet days"). Thresholds were calculated for each model ensemble member, with the ensemble mean threshold (Q95) used to estimate the future number of days per year (exceeding the threshold N95) and total annual rainfall from those days (P95). Q95 was not re-evaluated for the future climatological period.

Runoff was aggregated over each HUC2 watershed and multiplied by the respective area of to generate total volume per day. Volume per day was then converted to measurements more familiar to users, such as acre feet per day or cubic meters per second. Daily time series of total volumetric runoff had a 7-day running mean smoother applied, then annual maximum, minimum and mean values were extracted. The highest and lowest 7-day average runoff expected once per decade (7Q90, 7Q10) were estimated from twenty five years of annual maxima and minima.

## 4    Model Evaluation

The metrics used to evaluate CESM2's ability to reproduce large scale features and physical behaviors (e.g., Danabasoglu and Lamarque, 2021 and the associated Special Issue) are not necessarily those employed by decision makers. ESMs are designed to represent large-scale atmospheric processes and fluxes not specific local responses (Gettelman and Rood, 2016), but this design assumption may not be sufficiently well communicated to decision makers. The purpose of our evaluation is to establish whether CESM2 output is also fit for local decision purposes, or if the breadth of information from ESM ensembles remains unsuitable for immediate use in targeted water management decisions.

### 4.1    Rainfall metrics

While broad spatial patterns of seasonal mean daily rainfall are reproduced well (Danabasoglu et al., 2020; Feng et al., 2020; Simpson et al., 2022), CESM2 fails to capture details over high topography, and overestimates summer precipitation where convective extremes dominate summer rainfall (Appendix B). The seasonal mean precipitation also fails to capture some important watershed-level processes, such as the seasonal variability in the number of days with precipitation and the associated intensity.

Estimates of mean annual rainfall on wet days, or wet day volume, are in broad agreement between Livneh and CESM2 output. Figure 3 shows an example of the mean number of wet days per month (NWD), and mean wet




day volume (WDV) averaged over the Mid Atlantic and Pacific Northwest. While CESM2 represents the NWD
annual cycle very well in regions such as California (Fig. 3a, 3c) and the Pacific Northwest (Fig. 3b, 3d), it does
not capture NWD in many central and snow dominated regions. This is likely due to the smoother topography of
CESM2 missing the influence of orographic uplift, and large spatial scale missing sub-grid scale convective
systems (e.g., over the Central Plains).

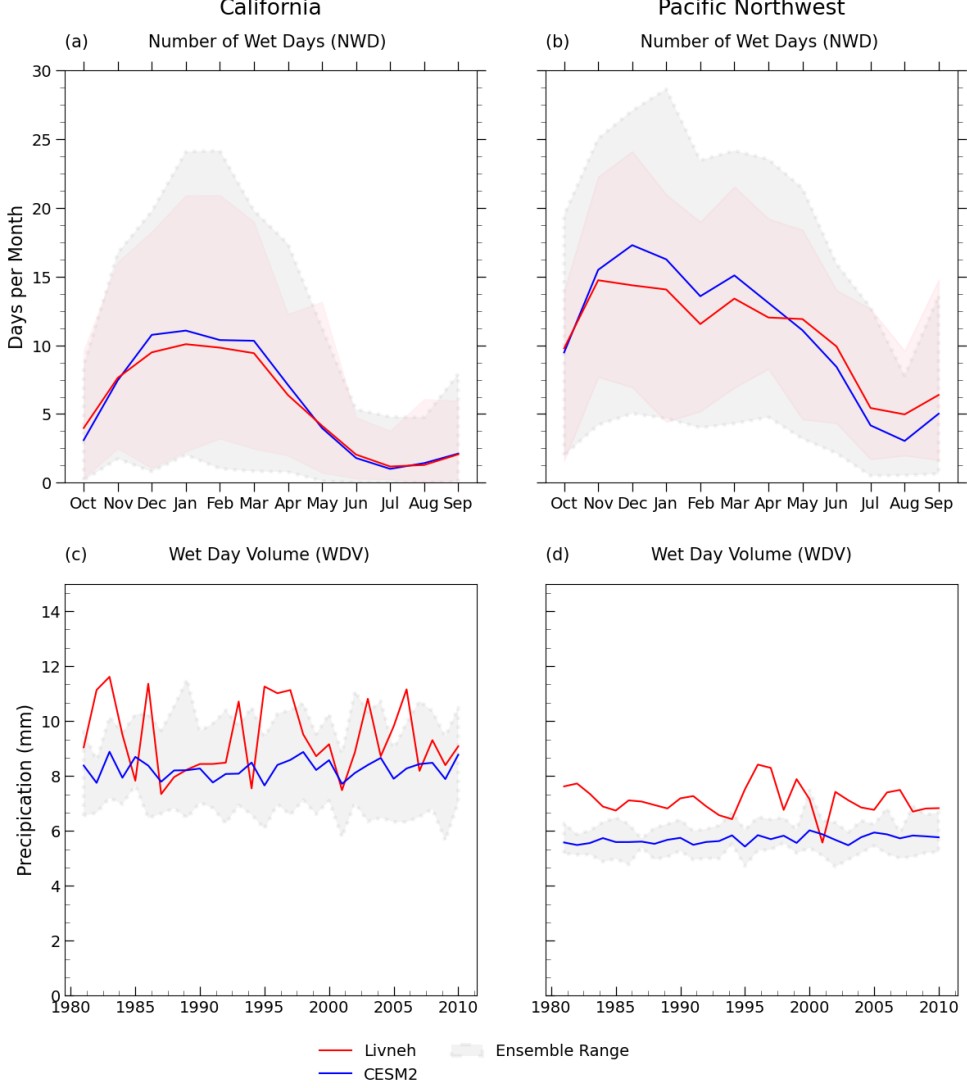

**Figure 3: Average number of wet days per month (a, b) and interannual variability in mean annual precipitation on
wet days for Livneh climatological mean (red) with interannual spread (pink) and CESM2 mean (blue) with
interannual and ensemble spread (gray); and (c,d) between 1981-2010 for observations derived from Livneh (red) and
CESM2 ensemble mean (blue) and spread (gray) in (a,c) Region 18 California (CA);
and (b,d) Region 17 Pacific Northwest (PN).**



The annual variability in WDV, both year-to-year variations as well as the overall range of minima and maxima, is well captured by each of the model members for the different HUC2 regions, even if the absolute values do not

match (Fig. 3 c,d). As expected, the specifics of which years have high or low values of WDV are not the same for each ensemble member (i.e. demonstrating internal variability). As a result, the ensemble mean value of WDV (blue) does not reflect the same year-to-year variability as the observations. Decision makers expressed that the interannual variability demonstrated by each model member is more valuable to demonstrate the credibility of the data than the ensemble mean (Tye, 2023). We recommend that the full range of values of each metric (i.e. after

computation for each ensemble member individually) are communicated in addition to the climatological means to help bound uncertainty around decisions (Wilby et al., 2021).

The magnitude of interannual variability in WDV (i.e., the absolute differences between the maximum and minimum values in each member time series) is typically within 10% of observations in all regions as illustrated

for two regions in Fig. 3. Exceptions are the Lower Colorado, South Atlantic-Gulf and Upper Mississippi where the simulated distributions are too narrow. While not as mountainous as, say, Upper Colorado these regions have a wide range of elevation changes not captured by the coarse model resolution that may contribute to the model-observation differences.

CESM2 captures the longest spells of consecutive dry days per year (CDD; Fig. 4a) and consecutive wet days per year (CWD; Fig. 4b), and their variability. Many regions capture both the interannual variability and the climatological mean duration of CWD, particularly those regions that are subject to large-scale synoptic systems (e.g., Pacific Northwest, Mid Atlantic-Gulf, California). Several regions either overestimate (South Atlantic-Gulf) or underestimate (Great Lakes, Souris-Red-Rainy) the absolute durations of the longest wet spells, but do reflect

the magnitude of interannual variability. The exception is Tennessee, where both interannual variability and mean CWD are overestimated. At the grid scale, broad spatial patterns of CWD are correct but the finer atmospheric processes arising from topographic features are incorrect, as expected from the coarse model resolution. A similar pattern is present in CDD, except that some drier regions with CDD >30 days do not capture the full range of interannual variability (Souris-Red Rainy, Missouri, Rio Grande). This is likely because all GCMs have a

tendency to produce drizzle (Vano et al., 2014); adjusting for a higher wet day threshold (e.g., 2 mm) might improve dry spell representation in those regions. It is also important to communicate such model sensitivities to users more effectively.





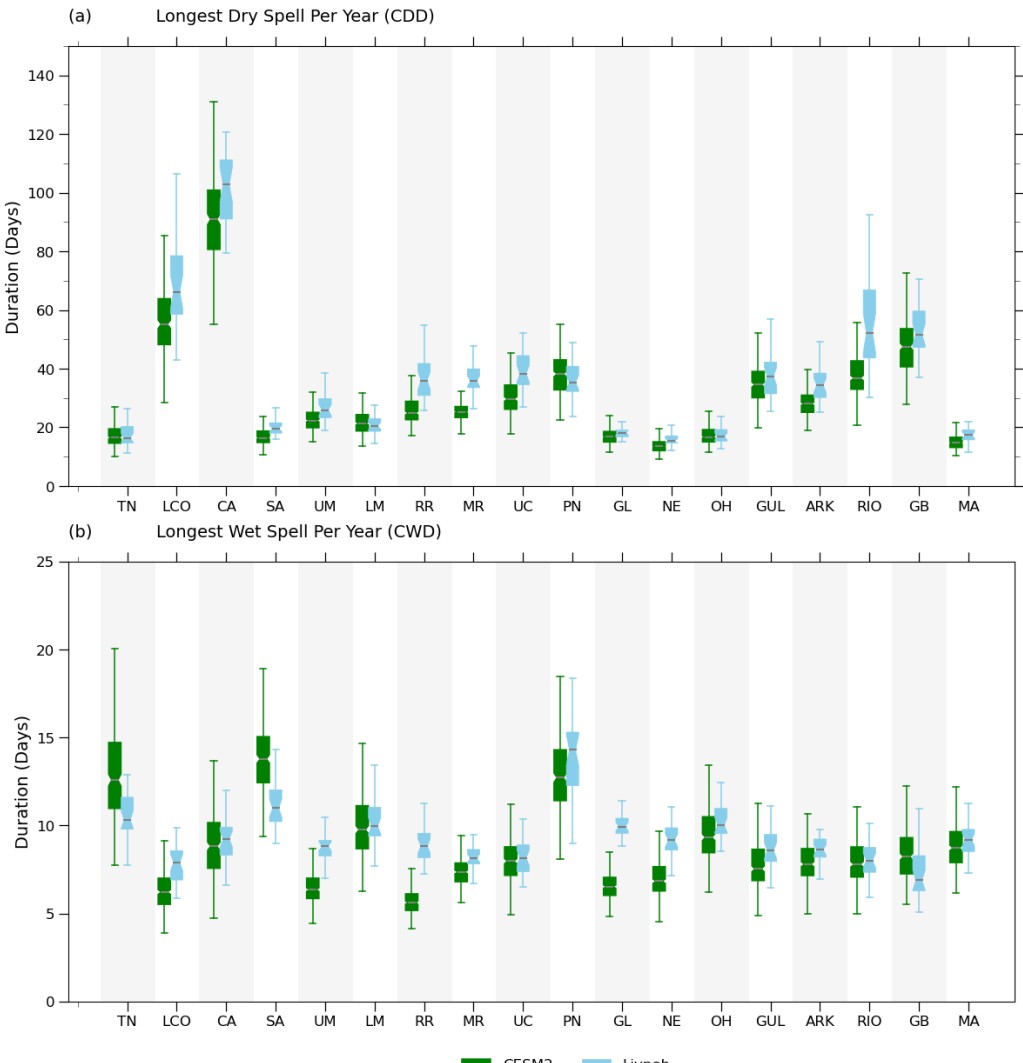

**Figure 4: a) Longest duration per year of consecutive days <1 mm rain (longest dry spell) for Livneh over all years (green) and CESM2 ensemble range over all years (blue) for all HUC2 regions; and b) Longest duration per year of consecutive days with ≥1 mm rain (longest wet spell). Regional Acronyms defined in Fig. 2.**

The thresholds for heavy and very heavy rain days (P95, P99) are defined individually for Livneh and CESM2 both to understand whether the intensity of more extreme rainfall is captured, and to evaluate model behavior. A comparison of the thresholds reflects the considerable improvements in modeling capabilities in recent years (Gettelman et al., 2022). For instance, earlier versions of CESM underestimated extreme precipitation intensity by 10-30 mm/day east of the Rockies, and overestimated intensity by 5-10 mm/day to the west (Gervais et al., 2014). We found CESM2 still underestimates the most extreme rainfall, but that errors have approximately halved. As these differences are still inadequate for many engineering and major infrastructure decisions (Wright et al.,



2019), we focus on CESM2's ability to capture the frequency of P95 and P99 per year, and their relative
contributions to the annual total. A result with considerable useability is the proportion of annual total precipitation
derived from the heaviest rain days, or "Proportional Contribution of Extreme Days" (P95Tot). This proportion
and its interannual variability is well represented by CESM2 at the HUC2 scale and has shown to be skillful in
other models (Tebaldi et al., 2021).

The interannual variability in the frequency (N95) and intensity of extreme rainfall, as represented by P95Tot, are
illustrated in Fig. 5. In several HUC2 regions the simulations report more frequent events, and proportionally
higher totals (e.g., Great Lakes, Rio-Grande, Missouri, Upper Colorado and Lower Colorado). Overall, there is
good agreement between Livneh and CESM2, identifying an opportunity to inform local decisions from large
scale ESMs.



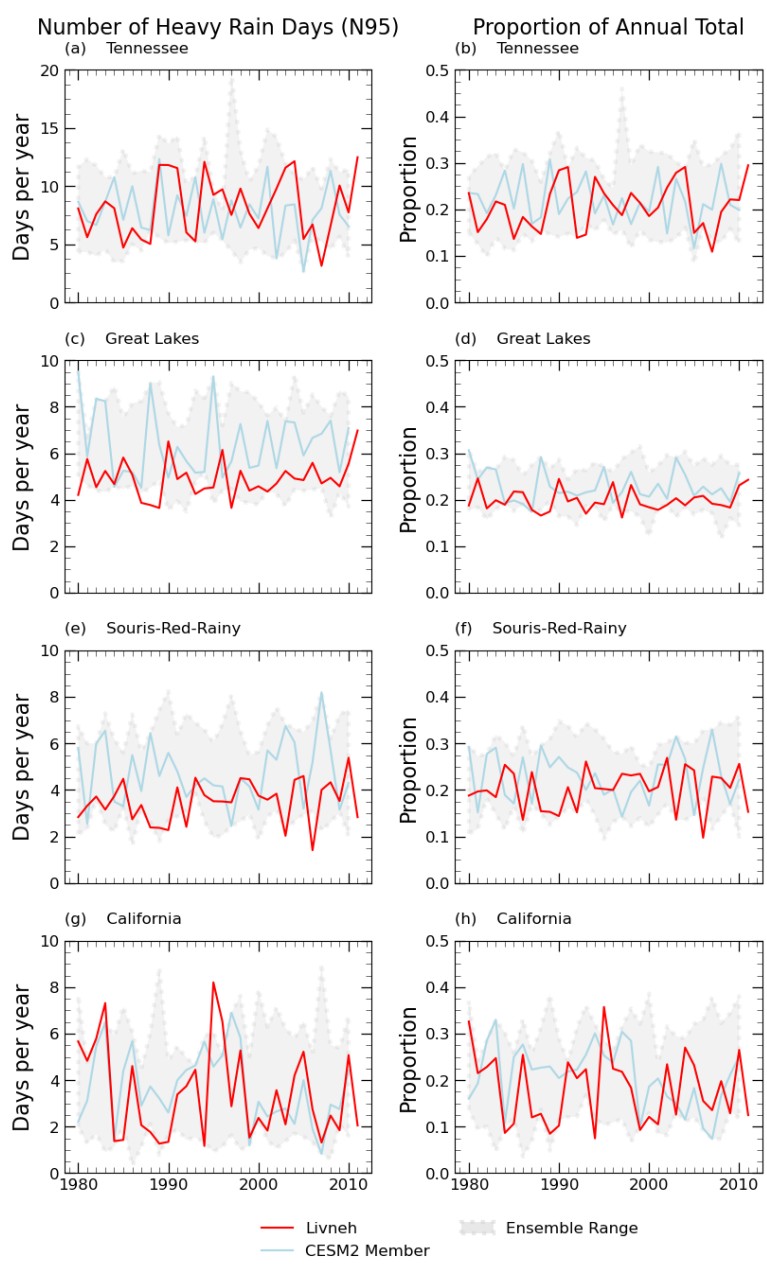


**Figure 5: a, c, e, g) Number of very heavy rain days per year; and b, d, f, h) total rain from very heavy rain days as a proportion of the annual total for a, b) Tennessee (TN); c,d) Great Lakes (GL); e,f) Souris-Red-Rainy (RR); and g,h) California (CA) HUC2 regions. Observations in red; CESM2 ensemble spread in gray, single ensemble member in blue.**




### 4.2 Runoff metrics

Runoff estimates are taken from the individual components of surface and subsurface runoff generated within CLM5 (Lawrence et al., 2019) and compared to the Livneh forced VIC runoff ("Livneh-VIC").

Assessing the skill of runoff in large-scale models is complicated by many factors, including the mismatch of scales between in-channel flow ($\sim$1-10$^2$ m) and the grid scale ($\sim$10$^5$ m). Thus, metrics of climate model runoff should be selected carefully and the runoff should be aggregated or combined with other metrics, rather than used directly (Lehner et al., 2019). Appendix C demonstrates the discrepancies between the grid-scale representation of runoff from Livneh-VIC and CESM2. The large discrepancies arise from different processes that are not

captured adequately, such as groundwater, topography, and associated snow ablation and melt, in addition to meteorological biases.

However, water management decisions are made over watersheds in units such as acre-feet[1] or cubic meters, while model data are output as a depth of runoff over each grid cell (e.g., mm/day per km$^2$). We aggregated the 7-day

running mean daily runoff (Q7) within each HUC2 region to generate Q7 time series in each basin. Fig. 6a illustrates the 25-year mean seasonal cycle for Livneh-VIC in red and CESM2 in blue, and the full range of values over all years and ensemble members for the Souris-Red-Rainy basin (HUC Region 9). Data are presented in millions of acre feet, to align with decision maker needs. The minimum simulated Q7 in any year considerably underestimates the lowest flows in this region compared to Livneh-VIC. In contrast, the largest total runoff volume

is overestimated and peaks too early in the water year. Figure 6b plots the same information as the cumulative runoff volume from the start of the water year, highlighting that the lowest runoff volume is underestimated by a factor of ten. Low runoff volumes were typically underestimated in smaller regions (e.g., NE, TN). High runoff volumes were only underestimated in three regions (LM, ARK, GUL) and considerably overestimated in seven regions. Snow-dominated regions perform particularly poorly for both QMax and QMin as snowpack and the

timing of associated runoff are not well simulated. Transitional regions that straddle both snow- and rain-dominated hydrology also fail to capture QMax, but better estimate Qmin (not shown). Only the South Atlantic region reproduces both QMax and QMin.

---

[1] 1 Acre-foot is the volume of water it would take to cover 1 acre of land to a depth of 1 foot. Equal to 325,852 gallons or 1,233 m$^3$ (USGS Water Science).



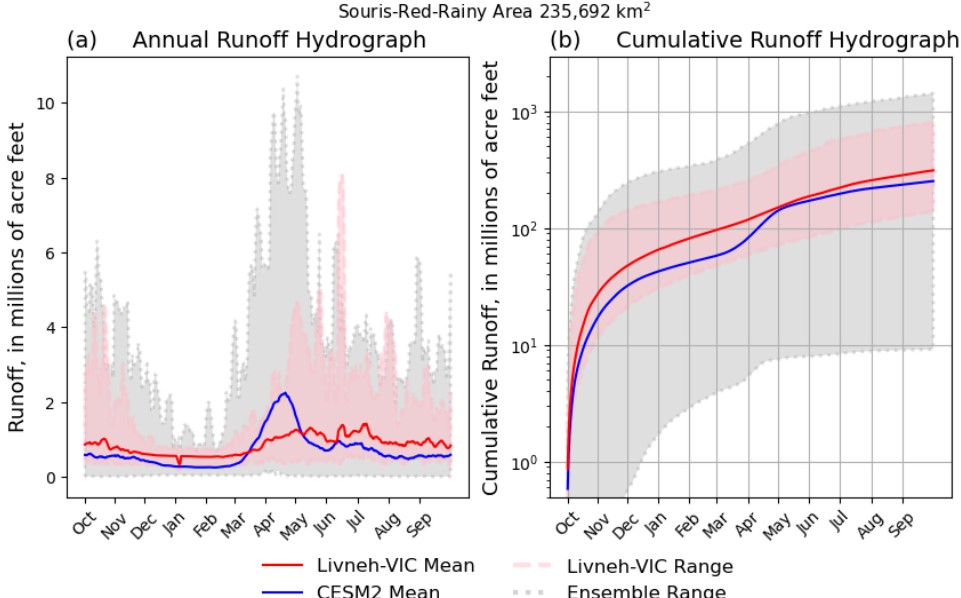

**Figure 6: Interannual variability in runoff in Souris Red Rainy Region for a) the mean seasonal cycle; and b) the cumulative watershed runoff over the water year. Livneh-VIC climatological mean in red, range of all years in pink; CESM2 ensemble mean in blue and ensemble range in gray. Figure highlights the underestimation of the lowest runoff volume by CESM2 by a factor of ten.**

We explored the relationship between the highest and total annual runoff (QMax/QTot), and lowest and total annual runoff (QMin/QTot). Some regions performed well for QMax/QTot, others performed better for QMin/QTot but there was no consistent relationship that could be utilized by decision makers.

Participants at the NCAR workshop (Tye, 2023) emphasized that the exact numbers produced by climate models are not very important for future decisions. Credible interannual variability and sensitivity to change signals are more important to give confidence in the direction of future changes (Lehner et al., 2019). Others have also emphasized the importance of well-represented processes in the model (Reed et al., 2022) and correlations with known experiences (Mach et al., 2020; Shepherd et al., 2018). Focussing on fidelity to the historical climate exaggerates the importance of model performance instead of robustness to different conditions without ensuring that model predictions are useful or reliable (Brunner et al., 2021; Wagener et al., 2022). Runoff estimates in transitional catchments may be inadequate in the current climate but plausible in the future, if the model reproduces rain-dominated hydrological processes (McMillan, 2021).

Climatological mean runoff cycles are estimated from Pardé coefficients — calculated as Q7/QTot on each calendar day — a dimensionless value that enables comparison across regions. Figure 7 depicts the mean seasonal cycle for representative snow-dominated (Upper Colorado), transitional (Missouri) and rain-dominated (Tennessee) regions, demonstrating how an imperfect representation of snow in the Upper Colorado results in





CESM2 peak runoff occurring two months earlier than Livneh-VIC (Fig. 7a). The runoff regimes display very different seasonal characteristics, with CESM2 having a "mid late spring" runoff regime rather than Livneh-VIC's "extreme early summer" regime (Fig. 7a; Haines et al., 1988). Peak runoff is also too early in transitional regions,

but closer to Livneh-VIC than in snow-dominated regions (Fig. 7b). Rain-dominated regions capture both the timing of QMax and overall seasonal hydrograph shape (Fig. 7c).

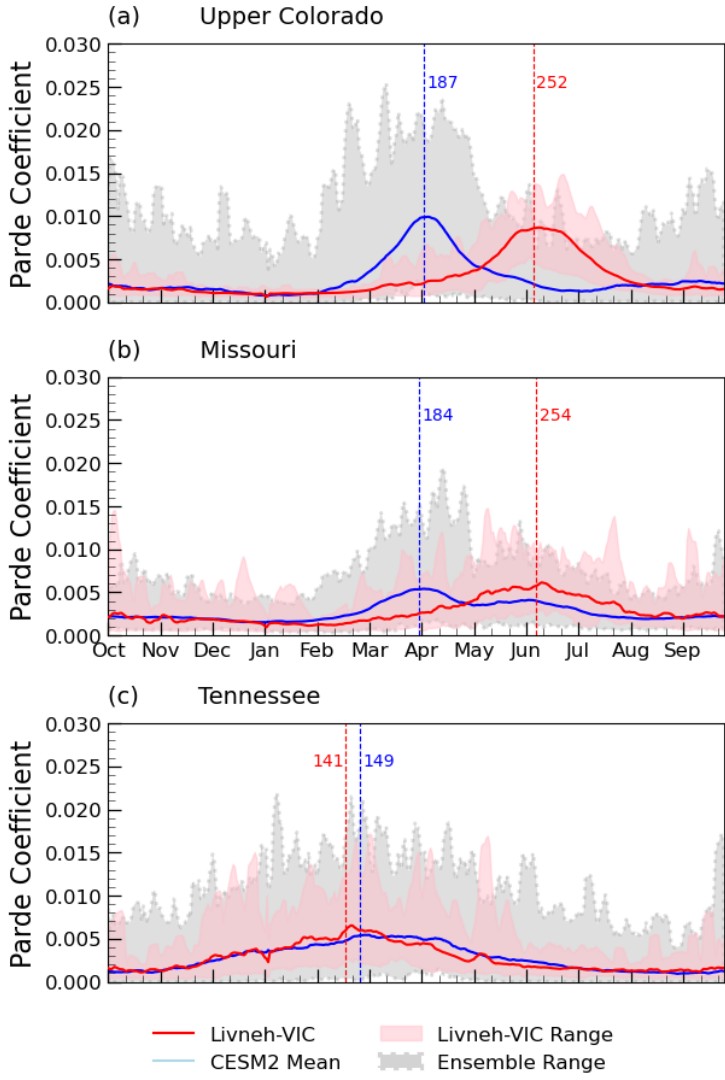

**Figure 7 : Seasonal patterns of runoff for HUC2 regions a) Upper Colorado (UC); b) Tennessee (TN); and c) Missouri (MR). Constructed from normalized series of the ratio of 7-day mean runoff to the mean annual total.**
**Livneh-VIC runoff climatological mean (red), climatological range (pink), CESM2 ensemble mean (blue) and ensemble range (gray with dashed border). Vertical lines indicate the mean date of peak runoff with number of days since the start of the water year.**



7Q10 and 7Q90 are estimated empirically from annual minima and maxima as occurring once per decade. Projected changes in the frequency of very low (high) runoff volumes are deemed credible where CESM2

replicates the standard deviation of annual minima and maxima according to a $\chi^2$ test at the 5% significance level. Table 1 reports CESM2 and Livneh-VIC regional estimates of 7Q10 and 7Q90 and standard deviation of the annual series; values in bold indicate where estimates are statistically similar.

**Table 1 : Very low (7Q10) and very high (7Q90) regional runoff, and standard deviation in regional annual minima ($\sigma$ QMin) and annual maxima ($\sigma$ QMax) for Livneh and CESM2. Values in bold indicate where CESM2 and Livneh-VIC regional runoff are statistically similar according to a $\chi^2$ test.**

| Region | | Livneh-VIC | | | | CESM2 | | | |
|---|---|---|---|---|---|---|---|---|---|
| | | **7Q10** | **7Q90** | **$\sigma$ QMin** | **$\sigma$ QMax** | **7Q10** | **7Q90** | **$\sigma$ QMin** | **$\sigma$ QMax** |
| NE | 1 | 4.1 | 132.4 | 1.3 | 25.5 | 8.6 | 215.1 | 4.7 | 39.9 |
| MA | 2 | **6.9** | 103.5 | **2.5** | 25.7 | **7.4** | 220.7 | **3.6** | 47.9 |
| SA | 3 | **21.1** | **240.4** | **8.4** | **50.7** | **20.5** | **258.6** | **11.9** | **45.8** |
| GL | 4 | **6.9** | 122.5 | **2.2** | 23.8 | **7.8** | 331.0 | **4.3** | 58.0 |
| OH | 5 | **7.8** | 187.6 | **2.3** | 53.0 | **9.4** | 260.9 | **4.5** | 56.4 |
| TN | 6 | 2.1 | **90.5** | 0.8 | **23.1** | 0 | **98.7** | 0.3 | **21.7** |
| UM | 7 | 2.1 | 78.2 | *1.7* | 16.9 | 7.9 | 122.3 | *4.7* | 31.5 |
| LM | 8 | 3.9 | 212.2 | 1.1 | 36.1 | 8.0 | 81.0 | 5.1 | 14.7 |
| RR | 9 | 1.0 | **24.3** | 0.5 | **7.1** | 0 | **33.0** | 0.1 | **8.4** |
| MR | 10 | 2.3 | 103.0 | 1.6 | 28.1 | 5.2 | 147.4 | 4.2 | 30.4 |
| ARK | 11 | 2.2 | 130.5 | 0.7 | 36.2 | 3.2 | 93.9 | 4.5 | 18.1 |
| GUL | 12 | 1.5 | 99.1 | 0.5 | 35.5 | 1.3 | 70.7 | 2.8 | 16.7 |
| RIO | 13 | **0.5** | **22.5** | **0.2** | **5.8** | **0.4** | **29.5** | **1.3** | **7.3** |
| UC | 14 | 0.6 | 27.3 | 0.2 | 7.2 | 0 | 74.7 | 0.2 | 15.3 |
| LCO | 15 | 0.5 | 19.4 | 0.2 | 7.5 | 0.3 | 46.7 | 0.7 | 11.6 |
| GB | 16 | 0.7 | 33.3 | 0.3 | 10.3 | 1.8 | 71.5 | 1.3 | 21.1 |
| PN | 17 | 20.6 | 266.5 | 7.9 | 50.2 | 4.4 | 449.6 | 2.6 | 87.3 |
| CA | 18 | 1.6 | 323.2 | 0.4 | 101.9 | 1.3 | 233.4 | 1.1 | 61.3 |

Grid-scale estimates such as mean daily runoff readily highlight why decision makers have low confidence in CESM2 output: the metrics are not salient and appear to have no skill. After aggregating the 7-day mean daily

runoff to watershed scales, some skill emerges in the annual minima and maxima, and seasonal cycles. Snow-



dominated watersheds perform poorly with regard to peak runoff volume and timing of the peaks and lows, as expected (McCrary et al., 2022). Rain-dominated watersheds capture the inter-annual variability and magnitudes of peak and low flows, and the seasonal hydrographs. While CESM2 at this coarse scale does not represent the local topography and cannot represent finer scale snow, our analysis indicates the land surface model correctly

simulates the bulk water budget. The projected runoff responses in regions that will have little to no snow in the future are, therefore, credible. This is premised on the understanding of *why* the model can produce accurate results, and whether the accuracy can be reliably reproduced for the future climate (Wagener et al., 2022).

## 5    Projected Changes

The analyses presented in Section 4 identified some rainfall and runoff metrics salient to water resource managers,
and well-capture by CESM2. While participants at the NCAR workshop stated that precise estimates are not necessary, they also emphasized their desire for high confidence in the projected scale and direction of any changes. We note that "confidence" is derived from a combination of 1) credible process representation; 2) agreement with historical trends, given internal variability; 3) agreement across multiple models. As the scope of this research was limited to testing the first aspect, we present projections for precipitation and runoff metrics in
the nine regions where CESM2 is credible.

### 5.1    Rainfall metrics

Projected precipitation metrics suggest no statistically-significant changes in the frequency of wet days (NWD) in any region by mid-century under the SSP2-4.5 emissions scenario. Mean seasonal precipitation is projected to increase in New England (NE) and Pacific Northwest (PN) during winter and spring, but overall changes are
slight. However, minor changes in the mean obscure the projected increasing intensity of the heaviest precipitation days and persistence of dry or wet spells (Donat et al., 2019).

Figure 9 compares the range of contributions to the annual total from very heavy rain days (P95Tot). The bars encapsulate the interquartile range of all years per region with black bars at the median of all years for Livneh
(blue) and all years and all ensemble members for CESM2 (green and orange); whiskers show the full extents of the data. Well-performing regions have the greatest overlaps between green (CESM2) and blue (Livneh) bars, while overlapping notches indicate statistical similarity. All regions show projected increases in the volume of annual total precipitation that will derive from the most extreme events, with significant changes indicated by divergent notches between green and orange (Future). Some regions (e.g., LCO, UM, GB) also show increasing
volatility of wetter or drier years, as indicated by longer whiskers and/or bars.

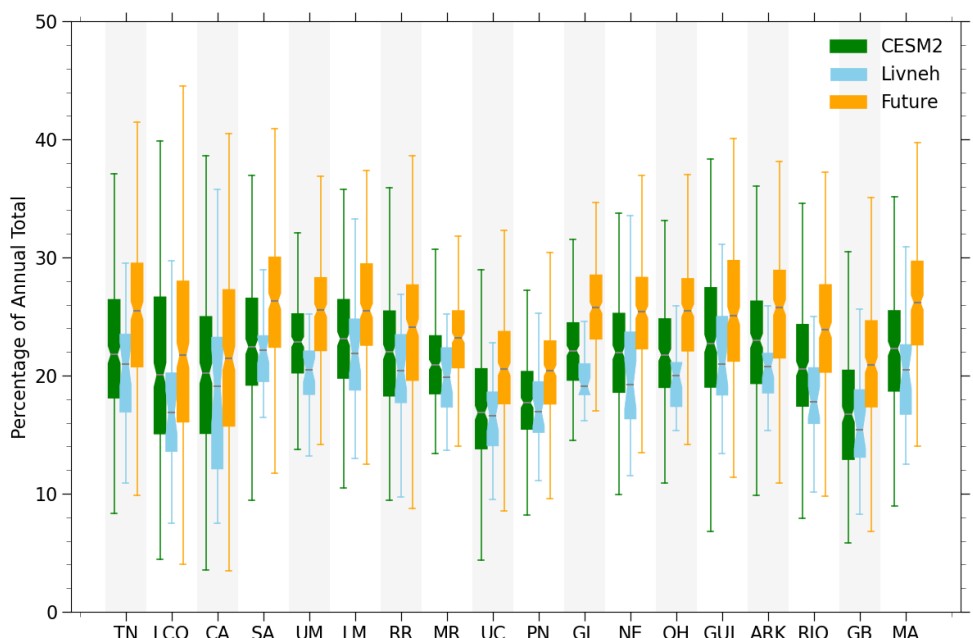

**Figure 8: Box plots of the interannual range of contributions to annual total rainfall from very heavy days (P95Tot) shown as percentages for: Livneh 1981-2010 (light blue), and also ensemble ranges for CESM2 1981-2010 (green) and CESM2 2040-2070 (orange) for all HUC2 regions. Boxes are bound by the interquartile range, black lines indicate the median, and bars extend to the full data range.**

Interannual variability is illustrated further for N95 (Fig. 9a) and P95Tot (Fig. 9b). Regions that are not statistically significant (for a student's t-test at 5% significance) are hatched. Both plots indicate the majority of basins in the west will experience a decrease (albeit statistically insignificant) in the interannual variability of N95 and their intensity. Great Basin (GB) is notable in projecting a significant increase in both the interannual variability in N95, and their intensity. This reduction in predictability could exacerbate existing water resource problems, and have potential consequences for the downstream basins (LCO, CA).





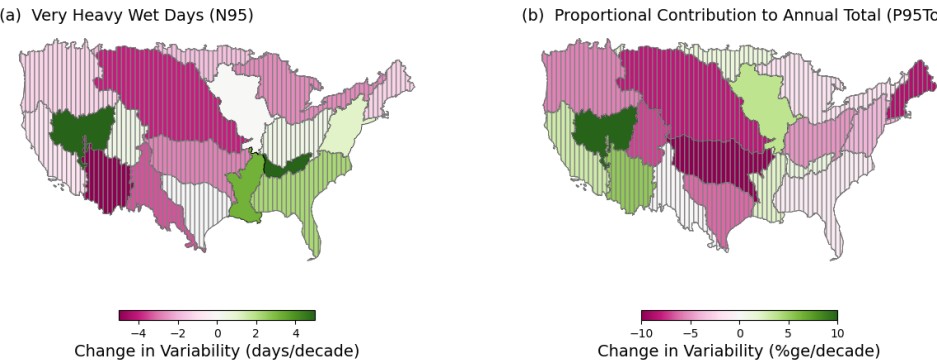

**Figure 9: CESM2 ensemble mean projected changes in interannual variability in a) frequency of very heavy wet days and units of days per decade; and b) proportional contribution to the annual total and units of percent per decade. Hatching indicates the region does not reach statistical significance.**

California is projected to halve the frequency of very heavy days, but the proportional contribution of those days to the annual total will increase from 20% to 22% (Fig. 8). This is coupled with projected increases in variability in the frequency and intensity of the heaviest events (Fig. 9), and reduced persistence in the duration of wet and dry spells (not shown). While not all of these changes are statistically significant, they are consistent with results from higher resolution models and suggest an increased potential for fire weather, drought, and floods (Lukas and Payton, 2020; Reclamation, 2016). Similar narratives are found for other regions, with several showing significant changes in the swings from wet to dry years (Fig. 9a). This emphasizes the importance of examining multiple precipitation metrics, and working with local partners to highlight potential risks and develop the full storyline of how future water management decisions relate to their experience.

### 5.2 Runoff metrics

CESM2-LENS projections could helpfully augment higher resolution model output in rain-dominated regions such as Tennessee, Ohio, and California, where CESM2 most closely reproduces Livneh-VIC. This is also true for transitional basins such as the Rio Grande, Northeast, and Lower Colorado, where seasonal snowpack may become more ephemeral and change the seasonal hydrological responses.

Based on the mean day of QMax, identified from Pardé coefficients, CESM2 projects QMax will occur around 5 days earlier in Tennessee, Ohio, and California by 2070 (Fig. 10). The duration of low flows at the end of the water year may also increase by around 5 days (not shown), but additional analysis using all CESM2-LENS members is needed to determine the true signal-to-noise in low-flow durations (Lehner et al., 2017). Transitional regions may experience QMax up to two weeks earlier as a result of changes in precipitation type.



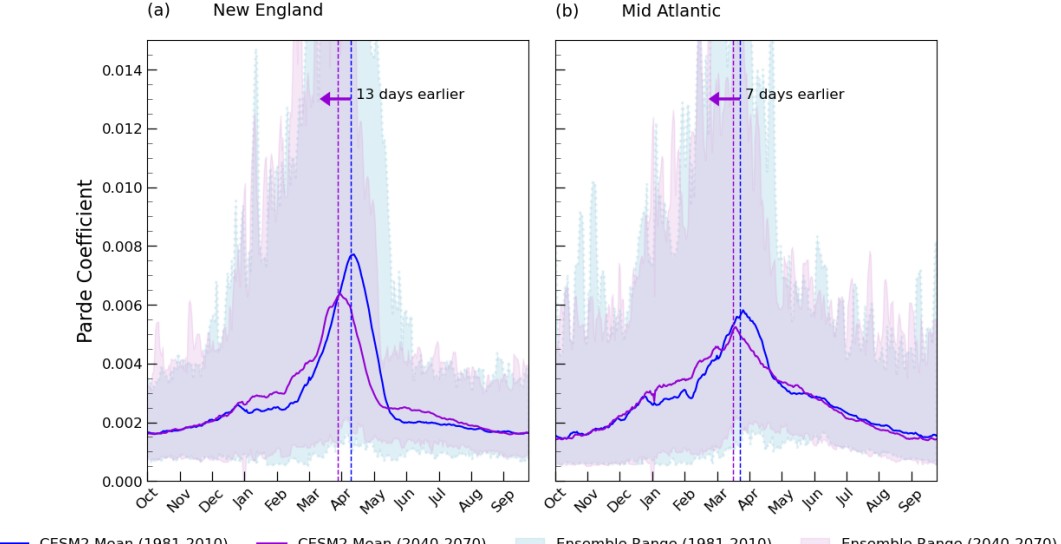

**Figure 10 : Example of projected changes in seasonal runoff timing for regions a) New England (NE);**
**and b) Mid-Atlantic (MA).**

The projected frequency of 7Q90 and 7Q10 has potential skill in CESM2 that would benefit water resource
managers. The projected changes in seasonal and multi-year behavior point to increases in the east-west divide in
drought-related problems. California is projected to have little change in 7Q90 frequency that may generate floods
but twice the frequency of very low events, while the South Atlantic may double or triple 7Q90 frequency with
little change in 7Q10 frequency. Table 2 compares the projected changes in frequencies of 7Q90 and 7Q10 events
between 2040-2070 and 1981-2005. Effective change is calculated from the difference in ensemble mean of the
expected rates over thirty years (i.e. 3 events in the current period). Color coding indicates a subjective human-
impacts assessment of beneficial (green) or adverse (orange) changes. Both 7Q90 and 7Q10 can have benefits
from an ecological perspective and so no change is the most beneficial condition. However, the built environment
is designed to be "fail-safe" (Tye et al., 2015) such that a lower probability of flooding would be beneficial, and
restrictions on water availability would be adverse.

**Table 2: Projected changes in the frequency of very high flows (7Q90) and very low flows (7Q10) per decade for well-**
**simulated regions. Color coding indicates beneficial (green) or adverse (orange) changes in runoff regimes**

| Region | 7Q90 per decade | Effective Change | 7Q10 per decade | Effective Change |
|--------|-----------------|------------------|-----------------|------------------|
| (1) NE | 0.4 | -1.0 | 1.1 | 0 |
| (10) MR | 0.2 | -1.0 | 2.0 | +1.0 |
| (18) CA | 0.8 | 0 | 2.4 | +1.0 |



| Region | 7Q90 per decade | Effective Change | 7Q10 per decade | Effective Change |
|--------|-----------------|------------------|-----------------|------------------|
| (2) MA | 0.6 | 0 | 1.2 | 0 |
| (3) SA | 3.3 | +2.0 | 1.0 | 0 |
| (5) OH | 1.0 | 0 | 0.8 | 0 |
| (6) TN | 2.0 | +1.0 | 2.0 | +1.0 |
| (9) RR | 1.0 | 0 | 3.4 | +2.0 |

## 6      Discussion

As decision makers have become more immersed in developing water resource management adaptation plans, the role of "climate services" in developing salient climate information has increased (Briley et al., 2020; Brugger et al., 2016; Dilling et al., 2019). We tested our hypothesis that recent improvements in ESMs can allow decision-relevant metrics to be produced directly, by leveraging the combined experience of the author team, results from the NCAR workshop, and the wealth of literature on actionable knowledge (Bremer et al., 2020; Jagannathan et al., 2021; Mach et al., 2020; Vano et al., 2014). Given that no model can perfectly address all decision needs, we identified and evaluated multiple metrics that can frame specific water management decisions within the known constraints of the data (Lempert, 2021), or within the decision makers' experiences (Austin, 2023; Clifford et al., 2020; Reed et al., 2022; Shepherd et al., 2018).

It is important to communicate the original purpose of the model and associated weaknesses, so that decision makers fully understand which information is appropriate to use in other applications (Fisher and Koven, 2020; Gettelman and Rood, 2016; Wagener et al., 2022). Given the balance between model fidelity and model complexity (Clark et al., 2015) and the absence of detailed global scale observation data (e.g., Gleason and Smith, 2014; Reba et al., 2011) CESM2 provides a plausible representation of Earth system processes and moisture fluxes, but may not capture basin-scale specifics (Ek, 2018; Lehner et al., 2019). That said, there are continued efforts to improve the simulation of land surface processes and analyses such as those presented in this article can flag weaknesses for future improvement (Lawrence et al., 2019).

Establishing model fidelity also requires distinguishing an accurate representation of the climate processes from serendipitous correlation with observations. Whether the model has good process representation overall, or exactitude in one simulation can be established through internal variability analyses using large ensembles (e.g., Deser et al., 2020; Tebaldi et al., 2021). Repeating the analyses with several different ESMs to establish the degree of agreement (Mankin et al., 2020) would further strengthen the usability of metrics presented in this article. It is also worth noting that the analysis presented here only used one reference dataset. As different reanalysis and observational datasets can have large discrepancies, a thorough model evaluation would also benefit from comparison to several products (Kim et al., 2020; Newman et al., 2015).



While the precise details of precipitation and runoff may not be well simulated by CESM2, we found some aspects
are sufficiently credible to support decision needs. The frequency of wet days highlighted regions where current
seasonal behavior is well captured, and future behavior is plausible enough to support planning around flood and
drought control or wildfire risk (Austin, 2023; Clifford et al., 2020; Jagannathan et al., 2021; Reclamation, 2016).
CESM2 projects increases in late spring and early fall rain, instead of snow, and in the longest wet and dry spells

affecting soil moisture capacity and the tendency for episodic floods and droughts in common with other basin-
wide assessments (e.g., Lukas and Payton, 2020; Underwood et al., 2018). Our analyses are also consistent with
higher-resolution model projections of increases in the most extreme rainfall events (Fowler et al., 2021).

## 7    Conclusions

This paper presented an assessment of whether a standard resolution (~100 km grid) Earth system model is capable
of producing information that water users typically employ in their decisions. Our motivation was to reduce the
need for intermediate downscaling and broaden the use of large model ensembles in localized decisions. We drew
on the combined experience of the project team and workshop participants to identify potential metrics and
familiar modes of visualization. This project used only CESM2 over the conterminous United States to develop
example metrics that may be explored within other models and over other regions.


Given the inherent limitations of large-scale models in replicating small-scale processes, we only presented future
projections for regions where processes are well-resolved on the coarse grid. We encourage others working in the
decision space between climate data producers and users to be forthcoming about specific regions and reasons
where model data are not credible, or where the model has particular weaknesses (such as the drizzle effect) that

may be overcome with a different analysis approach.
For future model assessors, the following metrics were found to be salient for water users and were skillfully
reproduced in many regions.

Rainfall:
▪    Number of wet days (≥ 1mm of rain) per year/season
       ▪    Mean precipitation on wet days
       ▪    Duration of the longest wet and dry spells per year
       ▪    Number of days with rain > 95th percentile of current climate wet day totals
       ▪    Proportion of the annual total derived from days > 95th percentile of wet day totals
Runoff (aggregated up to basin level, as a volume for 3- and 7-day averages):
       ▪    Annual maxima and minima
       ▪    Frequency of very high or very low flows (< 10% annual chance of occurring in the current climate)
       ▪    Proportion of averaged daily runoff to annual total



The work presented in this paper is a small step toward establishing greater usability of climate model output by decision makers. Continued collaboration is essential to improve the transfer of knowledge (e.g., data requirements, model assumptions, decision constraints) between communities.

**Appendix A**

**Table A1: Hydro-meteorological responses used in water management decisions, and the specific metrics that have potential for representation in ESMs. Metrics in bold are presented in this article.**

| Hydro-meteorological Responses | Typical Water Management Decision | Metric | Description |
|---|---|---|---|
| Annual rainfall | Water supply and drought monitoring | Total Precipitation (PRCPTOT) | Total annual precipitation measured as rainfall or snow water equivalent. |
| Seasonal rainfall cycle | Seasonal water supply, reservoir operations management | Number of Wet Days (NWD), Mean Wet Day Volume (WDV) | Frequency of days with ≥1mm precipitation (NWD) per month, season or year, Mean precipitation on wet days calculated from PRCPTOT/NWD |
| Rainfall extreme | Flood and stormwater management | 95th percentile (Q95) Number of very heavy rain days (N95) Very heavy rain volume (P95) Proportional contribution of very heavy rain (P95tot) | Rainfall percentile threshold that is exceeded by 5% rain events per year on average, and calculated from wet days only Frequency of days with rainfall exceeding Q95 Total rain falling on days exceeding Q95 Proportional of annual total derived from very heavy rain, calculated as P95/PRCPTOT |





| Hydro-meteorological Responses | Typical Water Management Decision | Metric | Description |
|---|---|---|---|
| Rainfall extreme (dry) | Water supply planning and drought monitoring/planning including water rights and restrictions. | Consecutive dry days (CDD) | Maximum duration of spell with consecutive days measuring < 1 mm precipitation. |
| Rainfall extreme (wet) | Stormwater management, water supply planning | Consecutive wet days (CWD) | Maximum duration of spell with consecutive days measuring ≥ 1 mm precipitation. |
| High streamflow | Reservoir management and flood control, water quality management and water supply management, including use of supplemental water supplies | Annual maximum runoff (QMax) Description (JMaxF) Description (HFD) | Annual maximum daily volume of basin-wide runoff Julian day of QMax/ day of the water year Duration of high flows |
| Low streamflow | Water supply management, assessment of water shortages with respect to seasonal demands | Annual minimum runoff (QMin) Description (JMinF) Description (LFD) | Annual minimum daily volume of basin-wide runoff Julian day of QMin/ day of the water year Duration of low flows |
| Streamflow | Water supply planning, water quality management, reservoir operations management, planning future investment needs | 7-day mean runoff (Q7) | Daily volume of basin-wide runoff averaged over 7 days. Often presented as percentage of annual total volume of runoff or Pardé coefficient (Pardé, 1933) |
| Very low streamflow | Water quality management for | 7-day "10-year" low runoff (7Q10)- | 7-day averaged basin-wide lowest volume of runoff with |



| Hydro-meteorological Responses | Typical Water Management Decision | Metric | Description |
|---|---|---|---|
| | discharge permits, conservation management, drought planning | | <10% annual probability of occurring. Estimated from Qmin series. |
| Very high flow | Flood management and planning, reservoir operations | 7-day "10-year" high runoff (7Q90) | 7-day averaged basin-wide highest volume of runoff with <10% annual probability of occurring. Estimated from Qmax series. |
| Streamflow | Water supply planning, reservoir operations management | Central Tendency (CT) Description ($Q_{25}$, $Q_{50}$, $Q_{75}$) | Day of the water year when the cumulative annual runoff exceeds 50% of the total annual runoff<br>Annual quartiles of cumulative annual runoff estimated from daily streamflow. |
| Snowpack | Reservoir operations and flood management, water supply planning | Snow Water Equivalent (SWE) Maximum (SWEMax)<br>SWEMax Date<br>SWE Duration | Volume of peak snow water equivalent<br>Day of the water year when peak SWE occurs<br>Total length of snow accumulation and ablation |
| Snowmelt | Flood management and reservoir operations | Snowmelt onset | Day of water year of snowmelt onset |



**Appendix B**

**Seasonal Mean Precipitation for Winter (top row), Spring (row 2), Summer (row 3) and Fall (bottom row) as shown in Livneh (left column) and CESM2 (middle column), and difference CESM2-Livneh (right column)**

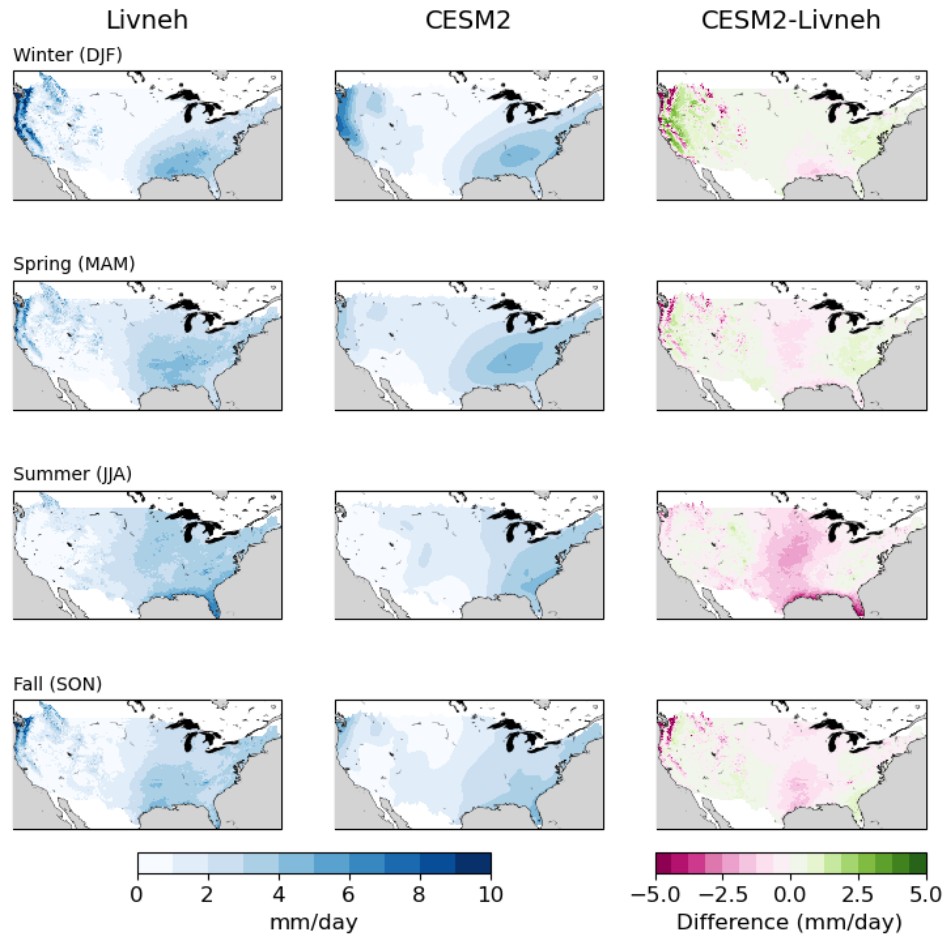



**Appendix C**

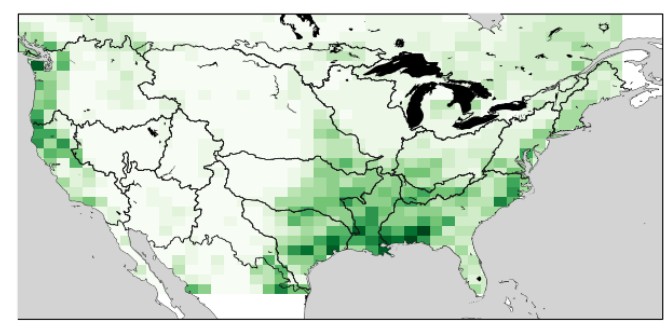

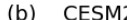

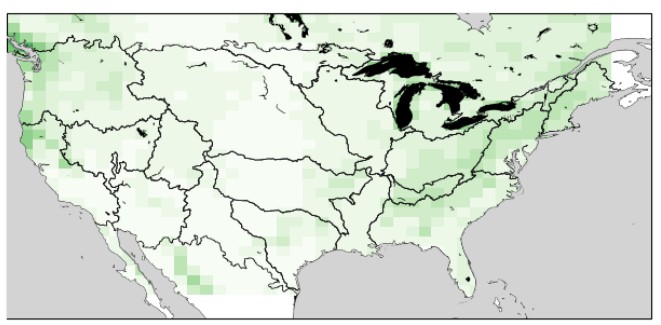

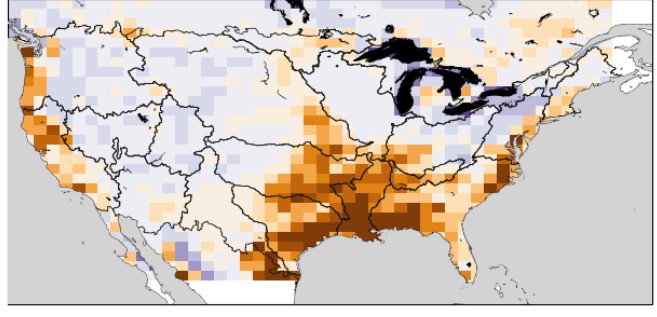

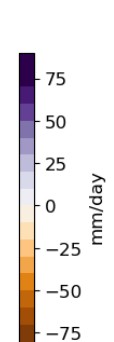



**Data availability**

All data generated for this study (e.g., CESM2 and Livneh-VIC calculated indices) along with Jupyter notebooks to recreate tables and figures are available in the repository https://github.com/maritye/PSIF_water_avail

**Author Contribution**

Conceptualization, M.T., J.R., E.G., A.N., A.W. and R.M.; Methodology, M.T., J.R., E.G.; Investigation, M.G.,
M.T.; Data Curation, M.G., M.T.; Writing - original draft, M.T., A.R., and R.M.; Writing – reviewing and editing, M.T,, J.R., E.G., A.N., A.W., R.M., A.R., F.L., C.B., and S.H.; Visualization, C.B., M.G. and M.T.; Supervision, J.R., E.G., A.N., F.L. and A.W.; Funding Acquisition, J.R., E.G., A.N., A.W., F.L.,C.B., S.H. and M.T.; Project Administration J.R.

**Competing Interests**

The authors declare that they have no conflict of interest.

**Acknowledgements**

This material is based upon work supported by the National Center for Atmospheric Research (NCAR), which is a major facility sponsored by the National Science Foundation (NSF) under Cooperative Agreement No. 1852977. Computing resources were provided by the Climate Simulation Laboratory at NCAR's Computational and
Information Systems Laboratory (CISL). The CESM project is supported primarily by NSF. We thank all the scientists, software engineers, and administrators who contributed to the development of CESM2. For the CESM2 Large Ensemble output we thank the CESM2 Large Ensemble Community Project and the supercomputing resources provided by the IBS Center for Climate Physics in South Korea. This research was primarily supported by the UCAR President's Strategic Initiative Fund. Portions of this study were supported by the Regional and
Global Model Analysis (RGMA) component of the Earth and Environmental System Modeling Program of the U.S. Department of Energy's Office of Biological & Environmental Research (BER) under Award Number DE-SC0022070.

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
