# Peer review of "Evaluating an Earth system model from a water manager"

_EGUsphere, 2023_

## Community Comment (CC1)

Sivarajah Mylevaganam

Alumnus, Spatial Sciences Laboratory, Texas A&M University, College Station, USA.

Often times, what we have is insufficient to meet what we need. There are times, what we have is more than what we need. Therefore, to bridge the gap that has been created between what we have and what we need, the principle of sustainable solutions has been preferred in the scientific field. This principle has been well accepted in the fields of economics and business development through the concepts of demand and supply curves.

In this manuscript, considering the spatial resolution and the obstacles introduced by global Earth System Models (ESM), the authors research whether what has been produced through ESMs is useful to meet every local-scale objective and need that is set by practitioners by means of metrics. The findings of the research reveal that some of the metrics that are set by local-scale objectives and needs are well produced by an ESM called the Community Earth System Model (CESM). Therefore, the authors request that we bridge the gap that has been created between what we have through ECMs and what we need through continued collaboration among all stakeholders.

1) The title of the manuscript needs to be evaluated by a specialist. In my opinion, these metrics that are evaluated against the model outcome are from practitioners and water managers. These metrics may not reflect what is expected from a water user. This could be well explained if we consider a river basin (e.g., the Mekong River Basin) that is pronounced for upstream-downstream conflicts. The metrics that would be desired by downstream users may not be favored by upstream users. Therefore, policymakers and practitioners decide metrics based on what is best to satisfy both parties (i.e., upstream and downstream users).

2) Line 85-88 (Given that ESMs have advanced immeasurably in the recent decade, it is time to re-evaluate whether their direct output can support decision maker)
It would be more appropriate for the authors to enumerate all the advancements in the model to understand these statements.

3) The table that has been presented in Appendix A is the culmination point of this research work .In my opinion, the authors need to add more information to understand the necessity of those metrics tabled by the practitioners. For example, as per the table, the number of wet days (NWD) is considered an important metric in reservoir operations management. The inclusion of an exact reason in this table would boost the contribution of this manuscript.

4) Refer to Appendix A
Mean precipitation on wet days calculated from PRCPTOT/NWD. Is this correct? As per the definition of PRCPTOT, it includes <1mm of precipitation as well.

5) Refer to Part II

**Acknowledgement and Disclaimer**

The author is an alumnus of Texas A&M University, Texas, USA. The views expressed here are solely those of the author in his private capacity and do not in any way represent the views of Texas A&M University, Texas, USA.

---

## Author Comment (AC1)

**Community Comment**

Sivarajah Mylevaganam

Alumnus, Spatial Sciences Laboratory, Texas A&M University, College Station, USA.

Often times, what we have is insufficient to meet what we need. There are times, what we have is more than what we need. Therefore, to bridge the gap that has been created between what we have and what we need, the principle of sustainable solutions has been preferred in the scientific field. This principle has been well accepted in the fields of economics and business development through the concepts of demand and supply curves.

In this manuscript, considering the spatial resolution and the obstacles introduced by global Earth System Models (ESM), the authors research whether what has been produced through ESMs is useful to meet every local-scale objective and need that is set by practitioners by means of metrics. The findings of the research reveal that some of the metrics that are set by local-scale objectives and needs are well produced by an ESM called the Community Earth System Model (CESM). Therefore, the authors request that we bridge the gap that has been created between what we have through ECMs and what we need through continued collaboration among all stakeholders.

> The title of the manuscript needs to be evaluated by a specialist. In my opinion, these metrics that are evaluated against the model outcome are from practitioners and water managers. These metrics may not reflect what is expected from a water user. This could be well explained if we consider a river basin (e.g., the Mekong River Basin) that is pronounced for upstream-downstream conflicts. The metrics that would be desired by downstream users may not be favored by upstream users. Therefore, policymakers and practitioners decide metrics based on what is best to satisfy both parties (i.e., upstream and downstream users).

This is a valid point, and we agree that the metrics developed are more useful for water managers of large basins rather than specific water users. We hope that this article will open the conversation for more researchers to explore where there is skill in ESMs that may supplement the information derived from higher resolution models.

> Line 85-88 (Given that ESMs have advanced immeasurably in the recent decade, it is time to re-evaluate whether their direct output can support decision maker)

> It would be more appropriate for the authors to enumerate all the advancements in the model to understand these statements.

There is a pretty vast literature on the advancements, some of which we have pointed out in the text.

The table that has been presented in Appendix A is the culmination point of this research work .In my opinion, the authors need to add more information to understand the necessity of those metrics tabled by the practitioners. For example, as per the table, the number of wet days (NWD) is considered an important metric in reservoir operations management. The inclusion of an exact reason in this table would boost the contribution of this manuscript.
Refer to Appendix A

There are no specific references for this, it stems from prior interactions with water managers. Interactions include the stakeholder workshop referenced in the text, as well as other collaborative projects such as described in Done et al. 2021

Done, James M., Rebecca E. Morss, Heather Lazrus, Erin Towler, Mari R. Tye, Ming Ge, Tapash Das, Armin Munévar, Joshua Hewitt, and Jennifer A. Hoeting. "Toward Usable Predictive Climate Information at Decadal Timescales." *One Earth* 4, no. 9 (September 2021): 1297–1309. https://doi.org/10.1016/j.oneear.2021.08.013.

Mean precipitation on wet days calculated from PRCPTOT/NWD. Is this correct? As per the definition of PRCPTOT, it includes <1mm of precipitation as  well.

Yes this is correct - it is the Standardized Daily Intensity Index defined by the ETCCDI (cited in the paper).

Refer to Part II

Acknowledgement and Disclaimer

The author is an alumnus of Texas A&M University, Texas, USA. The views expressed here are solely those of the author in his private capacity and do not in any way represent the views of Texas A&M University, Texas, USA.
Citation: https://doi.org/10.5194/egusphere-2023-2326-CC1

---

## Author Comment (AC2)

We would like to thank all three reviewers for their time taken to consider this manuscript and offering comments and constructive criticism. Responses to individual comments are included below adjacent to the reviewer comment.

A general point that was raised by all three reviewers related to the need for finer scale data to support more local decisions. The purpose of this research was not to remove the need for finer resolution data, rather to enable decision makers to benefit from the additional information they might obtain from using large ensembles of coarse resolution ESMs. As dynamical downscaling is computationally expensive, there are relatively few research centers able to produce a robust and coordinated multi-model regional climate ensemble that prioritizes the information needed by different model users (e.g. Goldenson et al. 2023) or single model initial-condition large ensemble of high-resolution models to quantify the internal variability of the regional model (e.g. Maher et al. 2021). While statistical downscaling is more computationally efficient, the resultant data would be enormous and so there is a similar lack of local scale multi-scenario-multi-model ensembles using statistical methods (e.g. Gutiérrez et al. 2019). Furthermore, in our experience many decision-makers do not have the time or resources to analyze such massive datasets. However, they do wish to understand how the local data they employ for decisions are situated within a cascade of uncertainty stemming from model and scenario selections (Wilby and Dessai, 2010). Our research premise was to understand whether any of the small-scale processes and extremes are plausible enough that the large ensemble uncertainty can be used to supplement higher resolution extremes, and reduce the burden on decision-makers. As a result we propose to remove the projected precipitation and runoff results, and associated information in the introduction and data, to improve the clarity of the message.

Goldenson, Naomi, L. Ruby Leung, Linda O. Mearns, David W. Pierce, Kevin A. Reed, Isla R. Simpson, Paul Ullrich, et al. "Use-Inspired, Process-Oriented GCM Selection: Prioritizing Models for Regional Dynamical Downscaling." *Bulletin of the American Meteorological Society* 104, no. 9 (September 22, 2023): E1619–29. https://doi.org/10.1175/BAMS-D-23-0100.1.

Gutiérrez, J. M., D Maraun, M Widmann, R Huth, E Hertig, R Benestad, O Roessler, et al. "An Intercomparison of a Large Ensemble of Statistical Downscaling Methods over Europe: Results from the VALUE Perfect Predictor Cross-Validation Experiment." *International Journal of Climatology* 39, no. 9 (July 2019): 3750–85. https://doi.org/10.1002/joc.5462.

Maher, Nicola, Sebastian Milinski, and Ralf Ludwig. "Large Ensemble Climate Model Simulations: Introduction, Overview, and Future Prospects for Utilising Multiple Types of Large Ensemble." *Earth System Dynamics* 12, no. 2 (April 22, 2021): 401–18. https://doi.org/10.5194/esd-12-401-2021.

Wilby, Robert L, and Suraje Dessai. "Robust Adaptation to Climate Change." *Weather* 65, no. 7 (June 2010): 180–85. https://doi.org/10.1002/wea.543.

**Reviewer Comment #1**

Citation: https://doi.org/10.5194/egusphere-2023-2326-RC1
The authors evaluate the performance of an earth system model (ESM), CESM2, in terms of a set of water availability metrics that support decision making. Here they focus on rainfall and runoff metrics. They found that, although the 100km resolution ESM may not match observations closely, it produces plausible and useful metrics for decision makers. This is from a very interesting perspective, i.e., from a water user perspective. However, the quality of the presentation needs to be improved.

As a person who is not familiar to CESM2, I would appreciate the authors could provide more information regarding the model, like a diagram of the model structure in the appendix, in addition to the sentences at the beginning of Section 3.1 and the reference to the model.

> Yes, we can include a diagram of the model structure in the appendix, e.g. Figure 1 from Danabasoglu et al. 2020
>
> > Danabasoglu, G., J.-F. Lamarque, J. Bacmeister, D. A. Bailey, A. K. DuVivier, J. Edwards, L. K. Emmons, et al. "The Community Earth System Model Version 2 (CESM2)." *Journal of Advances in Modeling Earth Systems* 12, no. 2 (February 2020). https://doi.org/10.1029/2019MS001916.

Since CESM2 still perform poorly on some metrics such as WDV and some regions including snow-dominated and mountainous regions, can the authors make some suggestions on how the model could be improved in the future?

> A companion paper by Rugg et al. (2023) examines potential improvements to the subgrid-scale simulation of land processes to improve the representation of the hydrological cycle in mountainous regions.
>
> Rugg, Allyson, Ethan D. Gutmann, Rachel R. McCrary, Flavio Lehner, Andrew J. Newman, Jadwiga H. Richter, Mari R. Tye, and Andrew W. Wood. "Mass-Conserving Downscaling of Climate Model Precipitation over Mountainous Terrain for Water Resource Applications." Geophysical Research Letters 50, no. 20 (2023): e2023GL105326. http://dx.doi.org/10.1029/2023GL105326.

Here the authors focus on the HUC2 regions. I am worried that the scale might be too big for local decision makers. Why not using a smaller HUC, e.g., HUC4?

While the scale of HUC2 regions may be large for some local decision-makers, it is also a more appropriate and conservative scale to compare to ESMs. Typically HUC4 and smaller watersheds do not contain many 100km$^2$ grid cells. Lehner et al. (2019) did carry out a comparison of runoff over smaller western US basins, demonstrating that caution is necessary in the direct use of climate model runoff, and the scale at which it is examined.

Lehner, Flavio, Andrew W. Wood, Julie A. Vano, David M. Lawrence, Martyn P. Clark, and Justin S. Mankin. "The Potential to Reduce Uncertainty in Regional Runoff Projections from Climate Models." *Nature Climate Change* 9, no. 12 (December 2019): 926–33. https://doi.org/10.1038/s41558-019-0639-x.

For evaluation, the authors use VIC outputs here, which are model results. Is there a plan to compare with in-situ runoff observations in the future?

A high-quality, long duration gridded database of unimpaired stream flow observations is not available. Runoff, as used in the paper, is not really observable. Another alternative would be the WaterWatch gridded runoff dataset, but this is also a simulated product. As such, we opted to use the VIC output produced by Livneh et al. (2015) as a quasi-observational set that covers the same time period as the observations and that was readily available for use.

Why do the authors use SSP2-4.5 here, not a SSP showing severe climate changes?

As noted in the general response, we will remove the future projections and references to SSPs to reduce confusion about the purpose of our research.

Specific comments:

Line 115: Which users here? Do the authors mean the model users?

This is a reference to the data users - it can be reworded as "water managers" to make the distinction.

Line 116: I think the bracket is in the wrong location, and it should be "(N95)".

Noted, thank you.

Line 213-215: It is hard to observe the similar annual variability in WDV between CESM2 and Livneh from Fig 3cd. Maybe show relative values.

We can add the variance in WDV in CESM2 and Livneh to plots 3cd.

This sentence will be re-written to emphasize how CESM2-LENS projections could provide supplementary information on the relative uncertainty in regional climate model output. Given that Livneh is on an approximately ~6 km grid, we estimate that the higher resolution models could be up to the order of 6 km$^2$ .

**Reviewer Comment 2**

Citation: https://doi.org/10.5194/egusphere-2023-2326-RC2

This paper provides a detailed evaluation of output from an earth system model, CESM2 over the conterminous US. The authors incorporate some feedback from climate data users to help determine metrics that might be useful, and then determine whether CESM2 can simulate the variables with enough accuracy to be useful without postprocessing of the output. Doing this sort of analysis with the interests of end users of the information at the center adds a useful perspective. While the research presented in this paper is interesting and generally well done, a couple of major shortcomings should be addressed. First, by validating CESM2 in many ways and finding some regions and climatic zones where the output reasonably represents observations, that seems like a verification that CESM2 is a strong candidate for downscaling (like the screening done by Goldenson et al., BAMS 2023, https://doi.org/10.1175/BAMS-D-23-0100.1). While I understand the point of the exercise is to demonstrate the potential value of ESM output without downscaling, if a downscaling method could more closely align output with observed metrics at a finer spatial scale while retaining the large-scale signal from the ESM, it is difficult to see why one would forego downscaling when serving data to stakeholders. The second shortcoming is that, while some skill is clear, especially in rain-dominated basins, it is not shown that broadly averaged statistics at the HUC-2 spatial scale would be actionable information for water managers.

Thank you for these observations. Our intention was to assess whether ESM output can be useful without downscaling, but do not intend a priori to advocate for this to occur. As noted to Reviewer 1, there is currently a trade-off in the use of model output: either sample model structural uncertainty and internal variability (via ESMs) or to have more realistic looking projections (via downscaling). It is not always clear which is a better approach for practitioners. While both communities have leaned toward improved realism via downscaling, it could be argued that this has occurred at the expense of sampling other sources of uncertainty. If ESMs can be used directly to inform some issues, it may open the door to a more comprehensive sampling of uncertainty by having comparative statistics available from the large ensemble or multi-model ensembles. As you allude to in point number 10, such an assessment can provide some verification that CESM2 is a candidate for downscaling.

In light of the reviewer comments, we have decided to remove the sections related to future projections as it appears that they do not substantially add to the value of the manuscript. Please also see the general comments.

Specific comments:

1. Line 24, the "watershed scale" is mentioned, but the scales used in this study would be more accurately described as "continental" or maybe "regional."

   This can be changed.

2. Section 3.2, the observations are the widely used Livneh data. While the potential to use other data sets is mentioned at the end of the paper, it should be noted that Pierce et al. (JHM 2021, https://doi.org/10.1175/JHM-D-20-0212.1) found biases in extremes in the Livneh data set that produce large discrepancies in extreme precipitation statistics and runoff. A revised version of the Livneh data set is available.

   Thank you for pointing out this additional reference and data set which were not available when we started the research  project. We do not anticipate re-running the analyses with the revised precipitation data as the hydrological component would then be out of sync with the observations. However, we can add text related to the differences in the extreme precipitation metrics and whether that supports our results or not.

3. Lines 174 and 183, For precipitation quantiles and 7Q10 and 7Q90 was a distribution assumed when calculating these probabilities?

   No these are all empirical quantiles. Per the definition of 7Q10 and 7Q90, the annual maxima were ranked and a threshold derived that equated to approximately one event every 10 years. A sentence will be added to the text to clarify this point.

4. Figures 3 and 5, the gray shading for all ensemble members is very faint, and invisible when printed. It would be better to use the slightly darker shading with the dashed boundaries, as in Figs 6 and 7. As noted in lines 217-220, decision-makers prefer to see individual ensemble members rather than the mean, so maybe that should be done in Fig 3 (as is already done in Fig 5).

   These changes can be made.

5. Lines 225-228, It is not clear why poor elevation representation at large scales has caused issues in interannual variability in the South Atlantic and Gulf basins while the mountainous Upper CO and other areas with greater elevation variation apparently have reasonable representation. A plot of elevation variability versus error might strengthen this argument.

   You are correct that there are a number of different reasons that could contribute to the poor representation of wet day volume. An inability to resolve convective precipitation is also a likely candidate, as is the drizzle effect mentioned on line 237. The sentence from 226-228 will be rephrased to encompass these other potential sources of error.

6. Line 240, a minor point is that the drizzle issue has been known for a long time, so citing an earlier reference (Dai, J. Climate, 2006, Chen et al., J. Climate, 1996, …) would acknowledge that.

    This will be changed.

7. Line 254, this sounds like circular logic, where you determine the precipitation that is exceeded 5% or 1% of the time, and then calculate the frequency of occurrence of those events. The contribution of those events to the annual total precipitation is more meaningful.

    The sentence will be reworded. However, we believe that it is useful to verify that the number of events producing the high totals, and their interannual variability, is approximately correct under current conditions. Any change in the frequency of occurrence of those events is also useful to understand in a changing climate, not just their overall contribution to the annual total precipitation.

8. Lines 283-284, the first sentence of the paragraph can be deleted. It is an ordinary change in units that does not need elaboration.

    Will be changed.

9. Lines 309-310, this largely repeats what was stated around line 218, and does not need to be included here.

    This sentence can be removed.

10. Lines 314-316, the argument that in some places the biases in CESM2 may decrease as the climate shifts does not seem like something that would be helpful to decision-makers. For example, if errors in CESM2 are due to failure to represent orographic effects, then a warming atmosphere could exacerbate them.

    Our results suggest that in the transitional regions there is sufficient precipitation but that it is simulated as rain rather than snow. You are correct that in highly elevated regions where orographic precipitation may not be simulated, the biases in CESM2 may not be stationary under climate change. We can add a caveat to the sentence to the effect that further investigation is warranted to understand the nature of the bias, whether the orographic effects have only affected the nature of the precipitation, or also its intensity. As you note, a bias that stems from a physical limitation in the model would preclude statistical downscaling or bias correction methods being applied to these results.

11. Line 334, please do not use parenthetical opposites like "low (high)" (Robock, 2010, doi:10.1029/2010EO450004)

    This sentence can be re-written.

12. Table 1: Why are some values in italics?

The italics will be removed.

13. Line 349, "our analysis indicates the land surface model correctly simulates the bulk water budget" is not clearly supported from Table 1. It appears only 4 or 5 (depending on the metric) of the 18 HUC-2 basins are statistically similar to the Livneh data. What is the threshold for being "correctly simulated"?

Table 1 illustrates the more extreme responses - i.e. one day maxima and minima with 10% or less chance happening in any year. As such they are not representative of the bulk water budget, but the tail of the distribution. A sentence can be added to clarify that Figures 6 and 7 are representative of the mean behavior, but that tail behavior shown in Table 2 is not well simulated.

14. Line 350, "The projected runoff responses in the regions that will have little to no snow in the future are, therefore, credible" is also not well supported. In fact, the following sentence admits that. As noted in comment 10 above, if errors are due to failure to simulate orographic precipitation because of poor terrain resolution in CESM2, then biases could plausibly increase in the future. The conclusion here seems to be that the runoff does not match Livneh in most locations and that more work is needed to determine why.

This is a valid conclusion. We will remove the sentence, and add a sentence indicating that runoff may be acceptable for some purposes, but that caution should be exercised for the reasons noted in your comment.

15. Line 360, which are the "nine regions where CESM2 is credible"? In Table 1 it looks like only two basins have statistically significant correspondence with the Livneh data for all metrics.

This section will be removed.

16. Lines 368-370 largely repeats the figure caption, so is not necessary.

This section will be removed.

17. Lines 382-383 also repeat the figure caption

This section will be removed.

18. Lines 396-397, "While not all of these changes are statistically significant, they are consistent with results…" Changes that are statistically insignificant are indistinguishable from noise, so should not be the basis for drawing conclusions. Restrict the comparison of trends to those locations where significant changes were found.

> This section will be removed. That said, it is worth highlighting trends that may emerge from the noise at some point in the future to alert water managers and others of the need to monitor such eventualities.

19. Lines 408-409 "CESM2 projects QMax will occur around 5 days earlier in …California by 2020". As an example of one of my more significant concerns noted in the summary above, it is worth referring to Stewart et al. (Climatic Change, 2004) to appreciate the wide variability in changes in runoff timing across California, both historically and for future projections. It is hard to imagine a water manager making much use of a single projection of a change for all of CA (or HUC 18), even if CESM2 simulates it with some skill. Is there evidence of managers taking action based on this scale of information?

> This section will be removed. You raise a valid point about the wide variability in precipitation patterns and runoff timing across such a large and varied basin as CA. We can add a sentence to the discussion section regarding the importance of detecting whether skill is a side benefit of the mean across a very large spatial scale, or if it is still detectable at representative climate scales (say north and south California, or the San Juan Valley and the Sierra Nevadas).

20. Table 2 and Figure 10, what is the statistical significance of the projected changes (relative to no change)?

> This section will be removed. However, we note that understanding whether a tipping point or impactful change in regime is being approached is worthy of monitoring even if it is not statistically significant.

21. Lines 461-462, that some aspects of CESM2 precipitation and runoff are "sufficiently credible to support decision needs" or that these results are "plausible enough to support planning around flood and drought control…" was not convincingly demonstrated.

> This will be rephrased to explain that the results would be supplemental to higher resolution models. That is, having metrics that are aligned with output from RCMs could help to bound some of the uncertainty arising from selecting only one ESM to drive a single RCM by providing a measure of the internal variability.

Typos:

Thank you, these will be addressed in the revised manuscript.

Line 100: "lead" should be "led"
Line 133: "In part…" is a fragment that should be connected to the prior sentence.
Line 179, "area of to" is an error
Line 355, "well-capture" should be "well-captured"

---

## Author Response (AR2)

Water Availability Review 2 Responses

We would like to thank the reviewers once again for their time taken to evaluate the revised manuscript. Our responses to their individual comments are included below.

Reviewer 1

1. In the Responses to Reviewers document, the authors clearly describe the motivation for the paper. Two very helpful sentences state the aim as "...to enable decision makers to benefit from the additional information they might obtain from using large ensembles of coarse resolution ESMs", clarifying that decision makers "wish to understand how the local [downscaled?] data they employ for decisions are situated within a cascade of uncertainty stemming from model and scenario selections."

Those clear phrases should be captured a little better in the manuscript. The last sentence of the abstract points toward this as a conclusion, but not as a driving motivation for the study, so that could be added to the abstract.

The third sentence of the abstract has been revised to read "However, the models are seldom evaluated for their ability to reproduce metrics that are important for practitioners, such that practitioners can situate higher-resolution model outputs within a cascade of uncertainty stemming from different models and scenarios."

Also, line 98 includes a third bullet (as a motivation for the study) as looking at projected changes, which the revised paper no longer includes. That can be replaced by something like one of the quoted phrases above.

The bullet has been removed and replaced with "the range of CESM2 structural uncertainty and internal variability for these metrics"

2. Lines 377-381 is a residual paragraph that had been part of a section that was deleted from the paper (prior section 5.2 on climate projections). This should be removed.

Thank you for noting this, the paragraph has been removed.

Reviewer 2

My main point is that instead of showing results for all basins, you have chosen some examples – which in general is fine. However, some parts of the results are based on data that is not presented in the manuscript (see specific in-line comments below). I suggest to include that data (e.g. in the Appendix if you don't want to blow up the Figures) and refer to those Figures in the Results.

Thank you for this suggestion. We have now included the following Supplementary figures and pointed to them within the main text

- Average number of wet days per month (Fig 3a) remaining 16 basins - S1
- Variability in mean annual rainfall on wet days (Fig 3b) remaining 16 basins - S2
- Number of heavy rain days and Proportion of Annual total from heavy rain days (Fig 5) for 4 additional basins - S3
- Annual Runoff (Fig 7a) for 8 additional basins - S4
- Cumulative Runoff Hydrographs (Fig. 7b) for 8 additional basins - S5

p.1 line 16: Explaining the term/job "water (resource) manager" and what the respective responsibilities are might help for readers from different fields.

To minimize confusion, we have removed the first instance of "water resource manager", and similarly changed "water resource metrics" to "water management metrics" in the abstract to make the text more consistent with the article title.

p.3 line 98: I suggest to remove this motivation point ("how such metrics are projected to change"), since apparently you removed that part of the analysis during the revision.

This has been removed, and an additional motivation point added as suggested by Reviewer 1.

p.5 line 150: You have added some text to accommodate a reviewer question, however this addition needs to be explained, in order to not confuse readers. → I suggest adding sth like "Pierce et al. (2021) provide an updated version of the Livneh dataset, by ... " to explain this.

This now reads: "Pierce et al. (2021) provided an update to the Livneh data set to address time adjustments that result in an underestimation of the most extreme daily precipitation totals and resultant runoff and flood potential (Pierce et al. 2021). However, as we are also interested other measures of in precipitation and in runoff minimat, we did not employ the updated gridded observations."

p.6 line 187: "expected once per decade" – is this still the case with the current rate of climate change – or has it changed already?

This is a good point. We have added a sentence "Stationarity was assumed over the climatological period for the purposes of these analyses, acknowledging that changes may have already occurred in the frequency of these events."

p.7 line 200: Should say "Appendix C".

Changed.

p.7 line 207-210: Why is there no figure for regions where there are larger differences? Would be interesting to compare. If the discrepancies are especially large in months with snow fall and or melt, I would assume that the missing processes related to snow accumulation and melt are playing a large role here.

This is a good point, we have added Supplemental Figure S1 for the other regions as noted above.

p.9 line 229-233: Again, here you are describing a results that you have not shown.

Supplemental Figure S2 has been added as noted above.

p.10 line 252: They are "defined" the same (both via the Q95 and Q99) I assume, but calculated separately and compared, correct?

This is correct. The sentence has been amended.

p.10 line 268-269: You write that the agreement is good (qualitative subjective judgement I guess), however the P95Tot and its variability is consistently overestimated by CESM2, compared to Livneh. You might want to mention that.

"Subjective" added to the sentence regarding good agreement. Supplemental Figure S3 has been added to illustrate the differences in other basins. However, we disagree that P95Tot and its variability is consistently overestimated - some basins are overestimated, others are well captured and this is already reported in the text.

p.12 line 287: Should say "Appendix D".

Changed.

Figure 3: Interannual spread for Livneh (pink area) is missing in the legend. You might want to use different shadings for the CESM2 range for the top (interannual + ensemble) and bottom panels (only ensemble), because they are not the same – and quickly mention this in the text to guide the reader.

Shading altered and legend updated.

Figure 5: Which single ensemble member do you plot and why? Why not plot the ensemble mean/median here?

The ensemble member was selected at random. This better shows the interannual variability, which is smoothed out and meaningless when the ensemble mean/median is used. This was in the text relating to Figure 3.

Table A1: "Metrics in bold are presented in this article." - I cannot find any bold text except for the headings.

The formatting has been amended to highlight the metrics used in the article in bold.

Reviewer 3

The authors present a work to evaluate Earth System Models by comparing its precipitation and runoff with that from gridded daily observations and VIC simulations. The motivation of this study is interesting to me, and can be helpful for the ESM users in hydrology. My major concerns come from the depth of the comparison. As we know, all the models may generate some results that can be comparable to but may be greatly different from the truth. A simple comparison cannot well address our concerns on their performance, especially for a water manager perspective, who may already has some information from remote sensing, in situ monitoring or other modeling.

This is a good point, we have added to the conclusions:

"The present evaluation is also only the first step in evaluating ESM performance. Additional research is needed to support water managers placing these results and their uncertainty in the context of additional observational data (such as remote sensing) that may already be available to them. "

(1) Only P and R are investigated, and how about other water balance compartments, such as soil moisture, evapotranspiration.

The reviewer makes a good point. At the beginning of this project, many water balance components were evaluated. In the interests of keeping this manuscript to a readable size we opted to focus on the most interesting results. As the available soil moisture and evaporation data were also from the derived Livneh VIC simulation, these results were not considered to present additional information for the benefit of the reader.

(2) The comparison between ESM and Livensh are displayed in a statistical way for some selected regions. How about direct validation at grid scale or in situ scale? Such information (or may be some cases) can better show the performance of ESM.

Direct validation at grid scale isn't appropriate for such a coarse scale, this is explained within the text and demonstrated in the figures in Appendix C seasonal mean precipitation and Appendix D Maximum daily runoff.

(3) The simulated runoff is used for comparison. Can in situ observations at river flow gauges be available for comparison? If so, the findings will be more reliable.

Again, comparing in situ observations to the coarse scale model results is not appropriate as described in the text. Furthermore, the results are of catchment runoff, rather than river flow and so a direct comparison cannot be made.

(4) Figure 3 shows the large deviations among different solutions in ESM. This large uncertainty should be highlighted for its use in practice.

Thank you for this suggestion, we have added a sentence at the end of the paragraph describing Figure 3:

The figures also highlight the scale of model (structural and internal variability) uncertainty present in the ensemble. As noted in previous sections, water management decision-makers are aware of the potential scale of uncertainty and expressed a desire for the full ensemble range to be presented to them instead of ensemble means.